# Evolution of Polymer Tantalum Capacitors

**Yuri Freeman** * and **Philip Lessner** *

KEMET Electronics Corporation, Simpsonville, SC 29681, USA
* Correspondence: YuriFreeman@kemet.com (Y.F.); PhilipLessner@kemet.com (P.L.)

**Abstract:** The major advantage of Polymer Tantalum capacitors over other types of tantalum capacitors is their low equivalent series resistance (ESR), providing a higher capacitance stability with frequency and ripple current capability. When Polymer Tantalum capacitors were introduced to the market in mid 1990s, they were low voltage, leaky, and unreliable, which limited their applications to commercial electronics. Today, some types of polymer tantalum capacitors demonstrate the highest working voltage, lowest DC leakage, and highest reliability ever achieved in tantalum capacitors. These Polymer Tantalum capacitors combine outstanding performance and reliability with superior volumetric charge efficiency, which makes them cost effective and attractive for numerous applications, including mission critical ones. This paper is dedicated to the major technological breakthroughs and scientific discoveries that enabled the radical evolution of Polymer Tantalum capacitors.

**Keywords:** tantalum capacitor; conducting polymer; working voltage; charge efficiency; reliability; cost

Polymer Tantalum capacitors with a sintered tantalum powder anode, an anodic oxide film of tantalum as the dielectric, and a polymer cathode were developed by NEC Corporation, Japan, and introduced to the market in mid 1990's [1]. The inherently conductive poly(3,4-ethylenedioxytheophene) (PEDOT) is the most commonly used cathode in polymer tantalum capacitors [2]. The conductivity of the PEDOT cathode is higher than the conductivity of the liquid electrolyte cathode in Wet Tantalum capacitors and the $MnO_2$ cathode in Solid Electrolytic Tantalum capacitors. This results in a lower equivalent series resistance (ESR), and thereby a higher capacitance stability with a frequency and ripple current capability in Polymer Tantalum capacitors in comparison with these characteristics in Wet and Solid Electrolytic Tantalum capacitors.

There were major improvements in the performance and reliability of Polymer Tantalum capacitors since they were introduced to the market, including a sharp increase in the working voltage and a decrease in DC leakage (Figure 1).

Figure 1 shows that original Polymer Tantalum capacitors had low working voltages and high DC leakage making them usable only in commercial electronics, while the current polymer tantalum capacitors have high working voltages and low DC leakage and are broadly used in high reliability applications [3].

There were major improvements in the technology of the Polymer Tantalum capacitors that enabled these radical changes in their performance and reliability. One of the most important improvements was flawless dielectric technology (F-Tech) [4]. The typical defects in the dielectric of tantalum capacitors are crystalline inclusions in the amorphous matrix of tantalum pentoxide ($Ta_2O_5$). These inclusions are initiated at the formation of the oxide dielectric, and continue to grow during the manufacturing, testing, and field application of tantalum capacitors. As the density of the crystalline $Ta_2O_5$ is higher than the density of the amorphous $Ta_2O_5$, the growth of the crystalline inclusions generates mechanical stress in the amorphous matrix that eventually causes cracks in the dielectric and failures of the capacitors. These crystalline inclusions start to grow from the small crystalline seeds at

the dielectric interface with the tantalum anode. Figure 2 shows the transmission electron microscopy (TEM) image and electron diffraction pattern of the $Ta_2O_5$ crystalline seeds on the surface of a tantalum anode after the amorphous matrix of the $Ta_2O_5$ dielectric was removed by etching in dilute hydrofluoric acid.

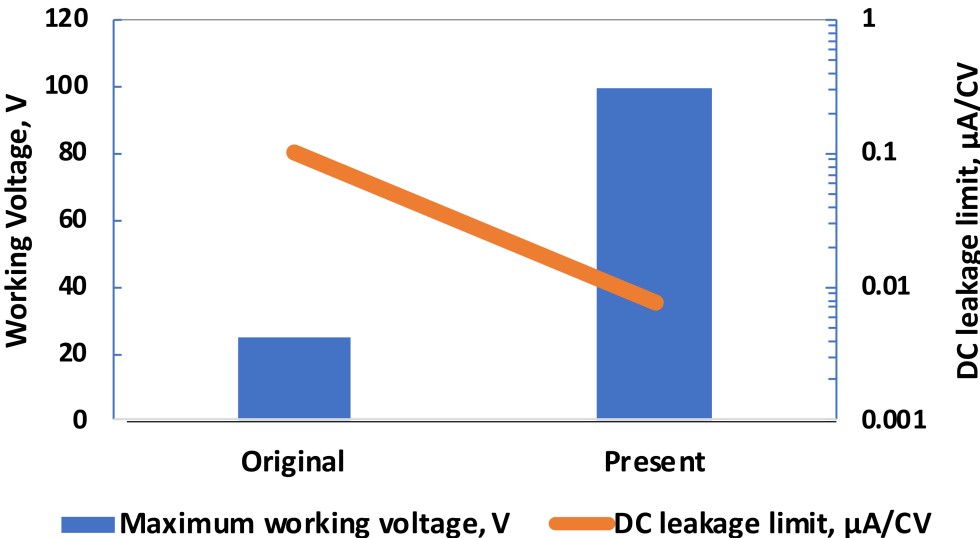

**Figure 1.** Maximum working voltage and DC leakage per CV limit in polymer tantalum capacitors.

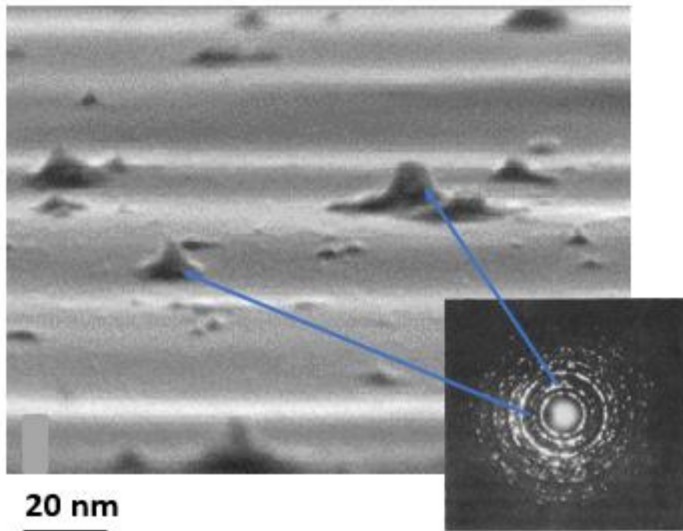

**Figure 2.** Crystalline seeds in the amorphous matrix of the $Ta_2O_5$ dielectric at the dielectric interface with a tantalum anode.

Figure 2 shows that crystalline seeds develop on the local sites in the tantalum anodes contaminated with impurities such as oxygen, carbon, and transition metals. F-Tech provides the highest chemical purity to the tantalum anodes, and thus eliminates the crystalline seeds and suppresses the crystallization of the amorphous matrix of the $Ta_2O_5$ dielectric.

There is special focus in F-Tech on reducing the oxygen content in the tantalum anodes, which increases during the vacuum sintering of the tantalum powder due to the diffusion of oxygen from the native oxide on the surface of tantalum particles into the bulk of the tantalum. Only sintering at temperatures equal or exceeding 1880 °C allows for oxygen to evaporate in a vacuum; however, most of the currently used tantalum powders would over shrink and totally lose their surface area at these high sintering temperatures. Reducing the oxygen content in tantalum anodes in F-Tech is typically performed by treatment of the vacuum sintered tantalum anodes in a deoxidizing atmosphere, such as

magnesium vapor, followed by leaching of the magnesium oxide in a diluted solution of sulfuric acid. Deoxidizing is performed at temperatures significantly lower than the sintering temperature, which allows for the preservation of the surface area of the sintered tantalum anodes.

Besides chemical purity, F-Tech also provides mechanical robustness to tantalum anodes, mainly through a strong mechanical bond between the sintered tantalum powder and the tantalum lead-wire that connects the anode with the external positive termination. In conventional technology, the tantalum lead-wire is inserted into the powder during pressing, and then the powder and the wire are sintered together at temperatures typically in the 1100–1600 °C range, depending on the powder charge and anode size. Mechanical stress during the manufacturing and testing of polymer tantalum capacitors can induce microscopic cracks in the oxide dielectric at the anode egress where the powder-wire bond is most vulnerable. These microscopic cracks may progress under additional stress during field application and cause catastrophic failures of the tantalum capacitors. In F-Tech, tantalum powder is pressed and sintered in a vacuum without the lead wire, and then the tantalum wire is resistance-welded to the anode surface in an inert atmosphere. The welding nugget with a strong powder-wire bond that forms at the anode egresses with this technology prevents the dielectric damage in this area under the mechanical stress.

In addition to the improvements in the chemical composition and mechanical robustness of the tantalum anodes, F-Tech also incorporates special anodization conditions, preventing overheating that can trigger the crystallization of the amorphous matrix of the dielectric [5].

Another important improvement in the technology of Polymer Tantalum capacitors that helped increase the working voltage and reduce the DC leakage was the process of applying a PEDOT cathode on the surface of the $Ta_2O_5$ dielectric. Originally, the PEDOT cathode was applied by multiple in-situ chemical reactions between the monomer and oxidant solution of iron (III) p-toluenesulfonate (FePTS). Although methanol and water washings were performed after every in-situ polymerization, it was practically impossible to fully wash out the by-products of the chemical reaction from the porous tantalum anodes. These byproducts, mainly iron, gradually accumulated in the PEDOT cathode and its interface with the $Ta_2O_5$ dielectric, pinning the potential barrier at the dielectric/polymer interface. In an alternative technology, tantalum anodes with the oxide dielectric were dipped in the water-based dispersion of the pre-polymerized PEDOT particles (slurry PEDOT), followed by drying in air at room temperature and elevated temperatures [6,7]. The slurry PEDOT technology does not have any byproducts that can contaminate the polymer cathode and its interface with the dielectric.

The combination of flawless dielectric and slurry PEDOT technologies in the manufacturing of polymer tantalum capacitors radically changed DC leakage/voltage and breakdown voltage (BDV)/dielectric thickness characteristics in comparison with these characteristics in the polymer tantalum capacitors manufactured with conventional dielectric technology and in-situ PEDOT (Figure 3).

Figure 3a shows the DC leakage increases rapidly with the voltage in the Polymer Tantalum capacitors manufactured with conventional dielectric technology and in-situ PEDOT, while it remains low in a broad range of voltages in the Polymer Tantalum capacitors manufactured with F-Tech and slurry PEDOT, providing sharp decrease in DC leakage. In Figure 3b, the BDV flattens at about 50 V with an increase in the dielectric thickness in the Polymer Tantalum capacitors manufactured with conventional dielectric technology and in-situ PEDOT, while it continuously increases with the dielectric thickness in the Polymer Tantalum capacitors manufactured with F-Tech and slurry PEDOT, providing a sharp increase in working voltages. A combination of flawless dielectric and slurry PEDOT technologies provide record low DC leakage and record high working voltage in the presently manufactured Polymer Tantalum capacitors [8].

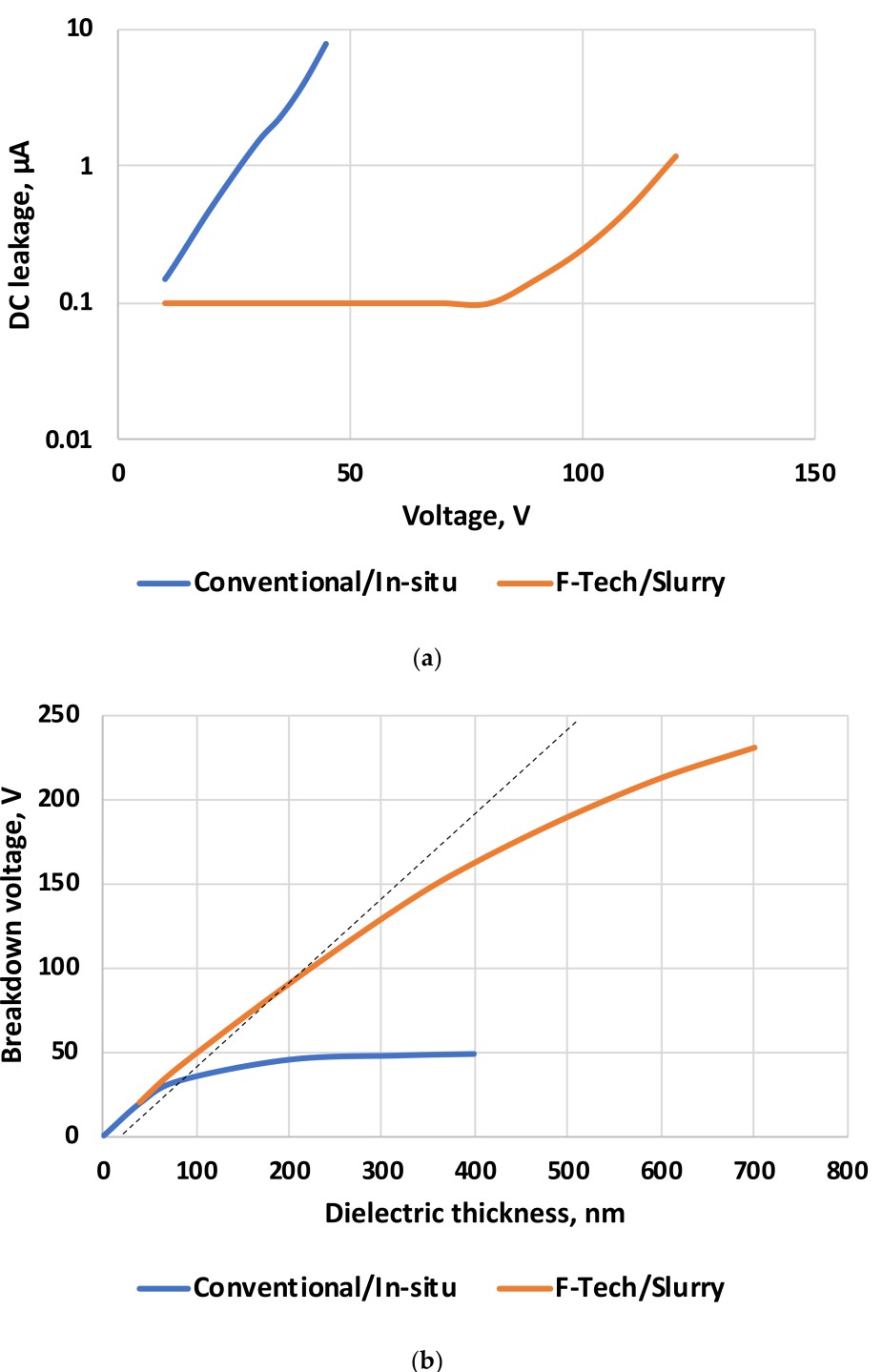

**Figure 3.** DC leakage/voltage characteristic with a 230 nm dielectric thickness (**a**) and breakdown voltage (BDV) dependence on the dielectric thickness (**b**) in Polymer Tantalum capacitors manufactured with conventional dielectric technology and in-situ PEDOT, and F-Tech and slurry PEDOT.

The potential barrier at the dielectric/polymer interface plays an important role in the differences in DC leakage/voltage and BDV-/dielectric thickness characteristics in Polymer Tantalum capacitors manufactured with either conventional dielectric technology and in-situ PEDOT or F-Tech and slurry PEDOT, as shown in Figure 3 [9,10]. This barrier increases with voltage at a normal polarity (plus on tantalum anode) and blocks DC leakage when F-Tech eliminates the defects that shunt the dielectric, and slurry PEDOT eliminates contamination of the polymer cathode that pins the potential barrier at the dielectric/polymer interface.

When polymer tantalum capacitors with a slurry PEDOT cathode (KV2 from Heraeus) are subjected to thorough drying, they exhibit anomalous charge current (ACC) exceeding the theoretical charge current ($I_{th}$), which is calculated as follows:

$$I_{th} = C * \frac{dV}{dt} \qquad (1)$$

where C is the capacitance and dV/dt is the voltage ramp.

ACC increases at higher applied voltages and lower ambient temperatures. In low voltage polymer tantalum capacitors with slurry PEDOT cathodes, the ACC is explained by the reorientation of the dipoles in charged polymer chains in the conductive polymer cathode at the dielectric/polymer interface [11]. This reorientation takes a longer time, with the slurry PEDOT having larger sized molecules of dopant poly(styrenesulfonate) (PSS) added to increase conductivity of the polymer cathode and to prevent the agglomeration of the pre-polymerized PEDOT particles in the water-based dispersion. Humidity in the polymer cathode plays the role of a plasticizer, reducing the reorientation time for the polymer chains and thus eliminating the ACC in the Polymer Tantalum capacitors.

F-Tech provides a significant reduction in ACC in higher voltage polymer tantalum capacitors with slurry PEDOT cathodes in comparison with the ACC with conventional dielectric technology. For example, Figure 4 shows the charge current distributions in D-case 15 µF–35 V Polymer Tantalum capacitors manufactured with slurry PEDOT cathodes and either conventional dielectric technology or F-Tech. The parts were baked out in the air at 125 °C for 16 h, cooled down to 0 °C, and charged immediately to 28 V with a ramp of 120 V/s.

As one can see in Figure 4, the average charge current in the Polymer Tantalum capacitors manufactured with conventional dielectric technology (Figure 4a) is about 10x the theoretical current $I_{th}$ = 1.8 mA calculated per (1), while it is about 2.5× the theoretical current in the polymer tantalum capacitors manufactured with F-Tech (Figure 4b). Continuous improvements in the structure and chemical composition of the tantalum oxide dielectric will further reduce the charge current in the Polymer Tantalum capacitors, with the slurry PEDOT cathodes eventually eliminating the ACC in these capacitors.

Polymer tantalum capacitors with lower working voltages are typically manufactured with a finer powder tantalum anode, thinner tantalum oxide dielectric, and hybrid in-situ PEDOT/slurry PEDOT cathode technology in order to cover the dielectric surface inside and outside the porous anodes. The F-Tech in the lower voltage polymer tantalum capacitors is based on the sintering of the pressed tantalum pellets in a deoxidizing atmosphere (deox-sintering) [4]. In this technology, a tantalum lead-wire is inserted into the pellet and is distorted during pressing at a high density in the region around the wire, providing a strong mechanical bond between the tantalum powder and lead-wire [12].

F-Tech with deox-sintering allows for combining a high reliability with record high volumetric charge efficiency CV/cc, the major advantage of tantalum capacitors in comparison with other types of the capacitors. Traditionally, high reliability capacitors are manufactured with a greater thickness of the dielectric for a given working voltage, and thus lose CV/cc as the capacitance is inversely proportional to the dielectric thickness. The formation of the thicker dielectric in tantalum capacitors requires larger sized powder particles in the tantalum anode, causing additional loss in CV/cc because of a reduction in the surface area of the tantalum anode. In polymer tantalum capacitors manufactured with F-Tech with deox-sintering, a high reliability is achieved without an increase in the thickness of the dielectric and related CV/cc loss.

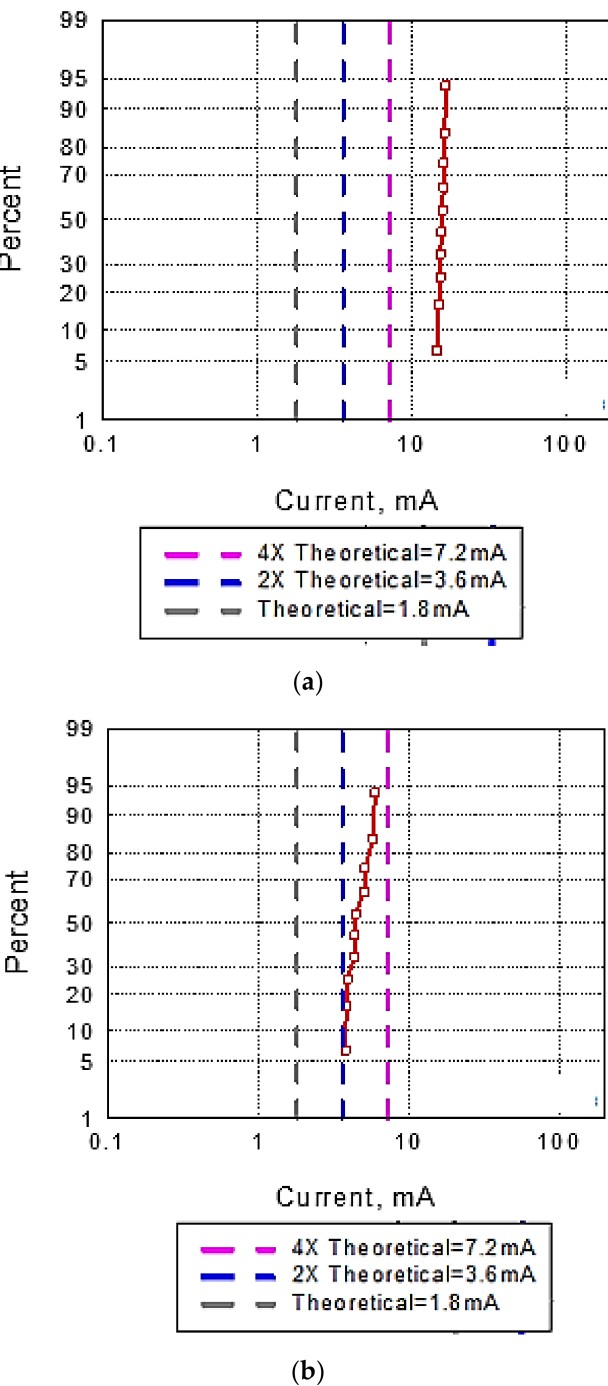

**Figure 4.** Probability plots for the current distributions in D-case 15 μF–35 V Polymer Tantalum capacitors with slurry PEDOT cathodes and either conventional dielectric technology (**a**) or F-Tech (**b**) charged to 28 V with a ramp of 120 V/s.

For example, Figure 5 shows the time-to-failure for accelerated tests at 105 °C and 1.3× working voltage and CV/cc in D-case 220 μF–16 V polymer tantalum capacitors manufactured with either conventional dielectric technology or F-Tech with deox-sintering. Both types of capacitors have similar thicknesses for the tantalum oxide dielectric and hybrid in-situ PEDOT/slurry PEDOT polymer cathode.

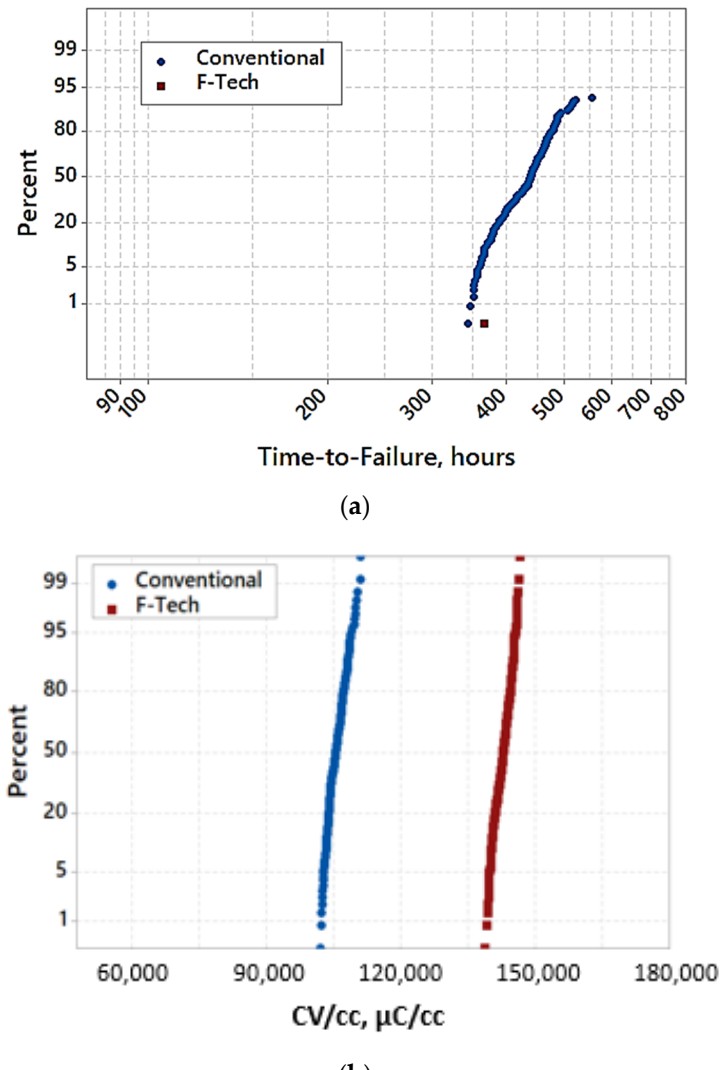

**Figure 5.** Probability plots for the cumulative percentage of failed versus time at accelerated testing at 105 °C and 20.8 V or 1.3× working voltage (**a**) and CV/cc (**b**) in D-case 220 μF–16 V Polymer Tantalum capacitors manufactured with either conventional dielectric technology or F-Tech with deox-sintering (200 parts with each technology).

According to Figure 5a, most of the Polymer Tantalum capacitors manufactured with conventional dielectric technology failed within relatively short time intervals (wear-out region), while only 1 out of 200 Polymer Tantalum capacitors manufactured with F-Tech with deox-sintering failed during this test. There were additional failures at higher accelerations and longer times, but no wear-out in the polymer tantalum capacitors manufactured with F-Tech with deox-sintering.

The high reliability of the Polymer Tantalum capacitors manufactured with F-Tech with deox-sintering is accompanied by about a 30% increase in CV/cc in these capacitors in comparison with the CV/cc in the polymer tantalum capacitors manufactured with conventional dielectric technology (Figure 5b). This increase in the CV/cc is due to the unique morphology combining large necks between the sintered powder particles with large pores in the tantalum anodes [4]. This morphology is practically impossible with traditional sintering in a vacuum, as thicker necks between the powder particles require higher sintering temperatures and thus more shrinkage and smaller pores in the tantalum anodes.

A high CV/cc and smaller capacitor size for the given capacitance and voltage correlates with a smaller size and lighter weight for the tantalum anode inside the capacitor,

which makes these capacitors cost effective, as the cost of tantalum constitutes a significant part of the total cost of the tantalum capacitor.

In conclusion, Polymer Tantalum capacitors underwent dramatic evolution from being low voltage, leaky, and unreliable when they were first introduced to the market, to the highest voltage, lowest DC leakage, and most reliable and efficient tantalum capacitors ever made. While applications of the original Polymer Tantalum capacitors were limited to commercial electronics, currently, these capacitors are broadly used in the most demanding mission critical applications. The major improvements in the performance and the reliability of Polymer Tantalum capacitors were achieved as a result of breakthroughs in their technology, including flawless dielectric technology (F-Tech) and pre-polymerized slurry PEDOT technology. These breakthroughs in technology were grounded on the scientific discoveries of the structural and chemical features of the basic layers and their interfaces in Polymer Tantalum capacitors.

**Author Contributions:** Both authors made equal contributions to all the aspects of the paper. All authors have read and agreed to the published version of the manuscript.

**Funding:** This research received no external funding.

**Institutional Review Board Statement:** Not applicable.

**Informed Consent Statement:** Not applicable.

**Data Availability Statement:** Not applicable.

**Acknowledgments:** Not applicable.

**Conflicts of Interest:** The authors declare no conflict of interests.

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
