# Peer review of "Evolution of Polymer Tantalum Capacitors"

_applsci, doi:10.3390/app11125514_

Round 1

Reviewer 1 Report

In this manuscript entitled “Evolution of Polymer Tantalum Capacitors”, the authors introduced two methods to improve the performance and reliability of tantalum capacitors: Flawless dielectric technology (F-Tech) and replacing in-situ synthesized PEDOT with pre-synthesized PEDOT ink. The topic fits the interest of Applied Sciences well and the results are promising. However, the manuscript is not flawless, and I would therefore recommend this manuscript to be published after major revision. Below please find the comments on this manuscript.

  1. Page1 line1, the authors named this manuscript “Evolution of Polymer Tantalum Capacitors”. I agree the results presented in this manuscript have a large impact on the community, but it is not proper to give a research manuscript this review-like title. A more specific title such as “the impact of defect control and conductive polymer preparation method on the performance of tantalum capacitors” would fit this research manuscript better.
  2. An experimental section needs to be added to introduce the materials and methodologies applied in this manuscript. The authors can also cite articles/patents that have been previously published.
  3. Page2 Figure. 1, the reference of original working voltage and DC leakage limit needs to be added in the figure caption.
  4. Page3 line 74, the sintering temperature needs to be specified and reference needs to be added.
  5. Page4 line 102, “doesn’t” needs to be spelled out as “does not”, similar for “it’s” (it is).
  6. Page5 line 123, the authors need to clarify which slurry PEDOT is applied in this manuscript. There are several commercially available PEDOTs on the market such as AI 4083 and PH1000, and their properties are significantly different from each other.
  7. Page7 line145, the function of PSS is serving as the counterion to balance the positive charges on the PEDOT chain and enable the water-dispersity of PEDOT:PSS compound. When comparing with other small counterion such as SO42-, PSS reduces the PEDOT conductivity due to its large size.
  8. Page8 Figure. 4, the authors need to describe the two figures are probability plots.
  9. Page10 Figure. 5, similarly, the authors need to describe the two figures are probability plots. Besides, in the first figure, do the red dots overlap with the blue dots? If so, the authors can consider making the red dots larger to help the readers understand.

I would recommend this manuscript to be published in Applied Sciences after the authors addressing the comments above.

Author Response

Reviewer1. Open Review

(x) I would not like to sign my review report
( ) I would like to sign my review repor11

English language and style

( ) Extensive editing of English language and style required
( ) Moderate English changes required
(x) English language and style are fine/minor spell check required
( ) I don't feel qualified to judge about the English language and style

Yes

Can be improved

Must be improved

Not applicable

Does the introduction provide sufficient background and include all relevant

 references?

(x)

( )

( )

( )

Is the research design appropriate?

(x)

( )

( )

( )

Are the methods adequately described?

( )

( )

(x)

( )

Are the results clearly presented?

(x)

( )

( )

( )

Are the conclusions supported by the results?

(x)

( )

( )

( )

Comments and Suggestions for Authors

In this manuscript entitled “Evolution of Polymer Tantalum Capacitors”, the authors introduced two methods to improve the performance and reliability of tantalum capacitors: Flawless dielectric technology (F-Tech) and replacing in-situ synthesized PEDOT with pre-synthesized PEDOT ink. The topic fits the interest of Applied Sciences well and the results are promising. However, the manuscript is not flawless, and I would therefore recommend this manuscript to be published after major revision. Below please find the comments on this manuscript.     Thanks to the reviewer for the positive review and good recommendations.

  1. Page1 line1, the authors named this manuscript “Evolution of Polymer Tantalum Capacitors”. I agree the results presented in this manuscript have a large impact on the community, but it is not proper to give a research manuscript this review-like title. A more specific title such as “the impact of defect control and conductive polymer preparation method on the performance of tantalum capacitors” would fit this research manuscript better.                                                            I wouldn’t change the title since in this short overview we focus only on two major breakthroughs in technology while many more were implemented to achieve radical improvement not only in performance but also reliability (Phil, I wouldn’t argue if you decide to change the title to follow the reviewer’s recommendation)
  2. An experimental section needs to be added to introduce the materials and methodologies applied in this manuscript. The authors can also cite articles/patents that have been previously published………………………………..The paper covers two technologies: F-Tech [4,5] and slurry PEDOT [6,7] where [4] is Y. Freeman’s book on Tantalum and Niobium-based capacitors published by Springer in 2017. The F-Tech is discussed in detail in chapter 3.3 of the book as a complex of special materials, processes, and testing techniques. The root causes of the typical defects such as crystalline inclusions in amorphous matrix of the dielectric are discussed in chapter 1 of the book.
  3. Page2 Figure. 1, the reference of original working voltage and DC leakage limit needs to be added in the figure caption…………………..Phil, please correct Fig. 1 (excel file attached)
  4. Page3 line 74, the sintering temperature needs to be specified and reference needs to be added………………………………….Sintering temperature for tantalum anodes is determined individually for each part-type depending of the type of the tantalum powder and thickness of the dielectric (chapter 2.1 in Freeman’s book). The optimal sintering temperature combines sufficient mechanical strength of the anode (achieved at higher sintering temperature and, thereby, larger shrinkage) while preserving surface area of the anode (achieved at lower sintering temperature and, thereby, less shrinkage)
  5. Page4 line 102, “doesn’t” needs to be spelled out as “does not”, similar for “it’s” (it is)……..Phil, please ask Rennie to correct this. She is very good with Word
  6. Page5 line 123, the authors need to clarify which slurry PEDOT is applied in this manuscript. There are several commercially available PEDOTs on the market such as AI 4083 and PH1000, and their properties are significantly different from each other…………..Phil, please answer this question
  7. Page7 line145, the function of PSS is serving as the counterion to balance the positive charges on the PEDOT chain and enable the water-dispersity of PEDOT:PSS compound. When comparing with other small counterion such as SO42-, PSS reduces the PEDOT conductivity due to its large size. ………………Reviewer is absolutely right. Small molecules of paratoluene sulfonic acid (pTSA) are used as PEDOT dopant in case of in-situ polymerization (chapter 2.3 in Freeman’s book). The problem with this technology is contamination of the cathode and its interface with the dielectric with byproducts of the in-situ chemical reaction. This contamination affects potential barrier at the dielectric-polymer interface and, thereby, performance of  the basic MIS structure of Polymer Tantalum capacitors where M stands for metal tantalum, I stands for insulator tantalum pentoxide, and S stands for p-type semiconductor PEDOT-dopant. Pre-polymerized slurry PEDOT/PSS technology addressed this issue by eliminating contamination at the dielectric-polymer interface.
  8. Page8 Figure. 4, the authors need to describe the two figures are probability plots.  The caption corrected in Fig.4 and Fig.5 - thanks to the reviewer
  9. Page10 Figure. 5, similarly, the authors need to describe the two figures are probability plots. Besides, in the first figure, do the red dots overlap with the blue dots? If so, the authors can consider making the red dots larger to help the readers understand…………..There is only one red dot in Fig. 5a as only one part with F-Tech failed during the testing while most of the parts with conventional technology failed at the same test conditions

I would recommend this manuscript to published in Applied Sciences after the authors addressing the comments above.

Reviewer 2 Report

The presented issue in the field of development of tantalum capacitors is an interesting issue showing the direction of their development. Unfortunately, the article is not suitable for printing in its current form. It shows the direction of development of KEMET tantalum capacitors, however, according to the reviewer, the material was presented very randomly and in a timely manner. What differs from the articles so far demonstrated by the authors, eg Environmental Stability of Polymer Tantalum Capacitors; E. N. Tarekegn, W. R. Harrell, I. Luzinov, P. Lessner, and Y. Freeman, Capacitance Stability in Polymer Tantalum Capacitors with PEDOT Counter Electrodes; Y. Freeman, I. Luzinov, R. Burtovyy, P. Lessner, W. R. Harrell, S. Chinnam and J. Qazi and a number of others in which co-authors of this publication appear. Min, these publications present similar information as in the article. Below is a summary of general and specific comments regarding the article:

1) The article seems to refer to the series of speeches by the co-authors concerning new polymero tantalum capacitors, but according to the reviewer, it is more information from some part of their research than presenting the evolution of tantalum capacitors.

2) References to the literature are inconsistent with those used in MDPI.

3) Why were new authors' works on tantalum capacitors omitted?

4) Fig. 1 incomprehensible, the leakage current is 0.005uA or 5 nA? or is it 0.005uA per 1V supply voltage? It is not so good that the scale on the right is linear and on the right is logarithmic.

What's on the x-axis between Orginal and Present? I would propose to give specific values ​​of leakage currents for old and new solutions.

5) Fig. 3a and Fig. 3b here would be the best logarithmic scale on the y axis which would show the superiority of the new solution (it would also explain the confusion related to Fig. 1). description point 3b unclear?.

6) Fig. 4 unclear red line is this a measurement?

7) Fig 5a would add that the tests were carried out on 200 pieces and gave the test voltage (probably 20.8 V because 1.3 x 16V), in the picture I would give the information that out of the tested 200 pieces only 1 was damaged.

8) General remark: the article requires significant editorial changes, the quality of the drawings is relatively low, one should consider the form of their presentation.

Author Response

Revier2.Open Review

English language and style

( ) Extensive editing of English language and style required
( ) Moderate English changes required
( ) English language and style are fine/minor spell check required
(x) I don't feel qualified to judge about the English language and style

Yes

Can be improved

Must be improved

Not applicable

Does the introduction provide sufficient background and include all relevant references?

( )

( )

(x)

( )

Is the research design appropriate?

( )

( )

( )

(x)

Are the methods adequately described?

( )

( )

(x)

( )

Are the results clearly presented?

( )

( )

(x)

( )

Are the conclusions supported by the results?

( )

( )

( )

(x)

Comments and Suggestions for Authors

The presented issue in the field of development of tantalum capacitors is an interesting issue showing the direction of their development. Unfortunately, the article is not suitable for printing in its current form. It shows the direction of development of KEMET tantalum capacitors, however, according to the reviewer, the material was presented very randomly and in a timely manner. What differs from the articles so far demonstrated by the authors, eg Environmental Stability of Polymer Tantalum Capacitors; E. N. Tarekegn, W. R. Harrell, I. Luzinov, P. Lessner, and Y. Freeman, Capacitance Stability in Polymer Tantalum Capacitors with PEDOT Counter Electrodes; Y. Freeman, I. Luzinov, R. Burtovyy, P. Lessner, W. R. Harrell, S. Chinnam and J. Qazi and a number of others in which co-authors of this publication appear. Min, these publications present similar information as in the article. Below is a summary of general and specific comments regarding the article: ………………………………………This paper was invited to the special issue just a few months ago and was written as a short overview of the major scientific and technological breakthroughs that provided drastic improvements in the performance and reliability of Polymer Tantalum capacitors. The paper is based on the authors’ earlier papers and Springer book while presenting principally important new experimental results. The paper is aiming not only engineers and scientists working with specific polymers, but much broader audience including developers and makers of Polymer Tantalum capacitors with inherently conductive polymer cathode as well as developers, makers and users of the end-electronic devices with these capacitors

1) The article seems to refer to the series of speeches by the co-authors concerning new polymero tantalum capacitors, but according to the reviewer, it is more information from some part of their research than presenting the evolution of tantalum capacitors…. Please see above

2) References to the literature are inconsistent with those used in MDPI…..Not sure what it is

3) Why were new authors' works on tantalum capacitors omitted? ……………….The new papers were not omitted, they were used as a scientific foundation for this paper (please see the initial comments.

4) Fig. 1 incomprehensible, the leakage current is 0.005uA or 5 nA? or is it 0.005uA per 1V supply voltage? It is not so good that the scale on the right is linear and on the right is logarithmic

What's on the x-axis between Orginal and Present? I would propose to give specific values ​​of leakage currents for old and new solutions. ….. Fig. 1 is corrected

5) Fig. 3a and Fig. 3b here would be the best logarithmic scale on the y axis which would show the superiority of the new solution (it would also explain the confusion related to Fig. 1). description point 3b unclear?.....                   Phil, the logarithmic scale of the y-axis in Fig. 3 doesn’t look good to me and I don’t understand what’s unclear with Fig. 3b.

6) Fig. 4 unclear red line is this a measurement?...... The captions in the Fig. 4 and Fig. 5 have been corrected.

7) Fig 5a would add that the tests were carried out on 200 pieces and gave the test voltage (probably 20.8 V because 1.3 x 16V), in the picture I would give the information that out of the tested 200 pieces only 1 was damaged…………. Requested info was added to the caption in Fig. 5

8) General remark: the article requires significant editorial changes, the quality of the drawings is relatively low, one should consider the form of their presentation.

Round 2

Reviewer 1 Report

The authors have fully resolved my concerns. The manuscript is recommended to be published in the current version. 

Reviewer 2 Report

Accept in present form